# Synthesis and Biological Evaluation of 1-(2-(6-Methoxynaphthalen-2-yl)-6-methylnicotinoyl)-4-Substituted Semicarbazides/Thiosemicarbazides as Anti-Tumor Nur77 Modulators

**DOI:** 10.3390/molecules27051698

**Published:** 2022-03-04

**Authors:** Hongyu Hu, Jiangang Huang, Yin Cao, Zhaolin Zhang, Fengming He, Xianfu Lin, Qi Wu, Shengxian Zhao

**Affiliations:** 1Department of Chemistry, Zhejiang University, Hangzhou 310027, China; huhongyu22@126.com (H.H.); xflin@zju.edu.cn (X.L.); 2College of Science and Technology, Ningbo University, Cixi 315302, China; 3School of Pharmaceutical Sciences, Xiamen University, South Xiang-An Road, Xiamen 361102, China; jxsdhjg@163.com (J.H.); 32320171153273@stu.xmu.edu.cn (Y.C.); fengminghe@stu.xmu.edu.cn (F.H.); 4Xingzhi College, Zhejiang Normal University, Lanxi 321004, China; ala_1208@163.com; 5Zhejiang Apeloa Kangyu Pharmaceutical Co., Ltd., Dongyang 322118, China

**Keywords:** semicarbazide, thiosemicarbazide, anti-tumor activity, Nur77, apoptosis

## Abstract

Nur77 is an orphan nuclear receptor that participates in the occurrence and development of a variety of tumors. Many agonists of Nur77 have been reported to have significant anticancer effects. Our previous studies have found that the introduction of bicyclic aromatic rings, such as naphthalyl and quinoline groups, into the *N*′-methylene position of indoles’ Nur77 modulators can effectively improve the anti-tumor activity of the target compounds. Following our previous studies, a series of novel 1-(2-(6-methoxynaphthalen-2-yl)-6-methylnicotinoyl)-4-substituted semicarbazide/thiosemicarbazide derivatives **9a**–**9w** were designed and synthesized in four steps from 6-methoxy-2-acetonaphthone and *N*-dimethylformamide dimethylacetal. All compounds were characterized by ^1^H-NMR, ^13^C-NMR and HRMS, and their anti-tumor activity on various cancer cell lines such as A549, HepG2, HGC-27, MCF-7 and HeLa are also evaluated. From the series of compounds, **9h** exhibited the most potent anti-proliferative activity against several cancer cells. Colony formation and cell cycle experiments showed that compound **9h** inhibited cell growth and arrested the cell cycle. Additionally, **9h** leads to the cleavage of PARP. We initially explored the mechanism of **9h**-induced apoptosis and found that compound **9h** can upregulate Nur77 expression and triggered Nur77 nuclear export, indicating the occurrence of Nur77-mediated apoptosis. These results suggested that **9h** may be a promising anti-tumor leading compound for the further research.

## 1. Introduction

Nuclear receptors are important targets for drug development, which are related to the therapeutic effects of 16% of small molecule drugs [1]. Nur77 is an orphan nuclear receptor in the nuclear receptor family. Although its natural ligand has not been discovered yet, it can be regulated by a variety of intracellular and extracellular stimuli and specific agonists/antagonists, and is involved in physiological regulation such as cell growth, differentiation, apoptosis, autophagy, metabolism, aging and immunity [2,3,4,5,6]. It has been found that Nur77 is a key participant and regulator in the occurrence and development of a variety of tumors [7], and it has abnormal expression in a variety of tumors, including liver cancer [8,9], prostate cancer [10] and colon cancer [11], and is closely related to tumor development, invasion and metastasis. Many agonists of Nur77 have been reported to have significant anticancer effects. For example, small molecule compound CsnB isolated from marine endophytic fungi [12], celastrol isolated from the root of Tripterygium wilfordii [13], sterol compounds ATE-I2-B4 and H-9 isolated from toad skin [14], ginsenoside compound CK isolated from ginseng [15] and bisindole compounds DiM-C-pPhOH [16] and shikonin derivative SK07 modulate the Nur77-Bcl-2 apoptotic pathway [17]. These studies suggested that Nur77 is an ideal and effective drug target for the development of anti-tumor small molecule drugs.

In recent years, our group has studied the anti-tumor activities of a series of indoles derivatives as novel Nur77 regulators on liver cancer cells, lung cancer cells, breast cancer cells and gastric cancer cells [18,19,20]. However, currently synthesized small molecule compounds targeting Nur77 are not strong enough in binding, activity and specificity. On the other hand, Nur77 plays a dual role in promoting the proliferation and death of tumor cells, and its biological function depends on different external stimuli and is tissue and cell specific [21]. Nur77 is involved in the regulation of various metabolic processes and its role and mechanism in tumor formation remain to be further clarified [21].

In our previous studies, it was found that indole-spliced urea derivatives can induce Nur77 expression and transfer from nuclear to mitochondria and activate the apoptosis pathway to inhibit the growth of gastric cancer cells [18]. Interestingly, we found that the introduction of bicyclic aromatic rings, such as naphthalyl and quinoline groups, into the *N*’-methylene position of indoles Nur77 modulators can effectively improve the anti-tumor activity of the target compounds [19]. Over the past few years, we found that some heterocyclic urea/thiourea derivatives have a wide range of anti-tumor activities (Figure 1) [18,22,23]. As part of our ongoing study, here we synthesized a series of naphthalene pyridine urea/thiourea derivatives, which exhibit significant inhibitory effects on a variety of tumor cells, especially in gastric cancer cells. Further studies showed that compound **9h** could inhibit the growth and proliferation of gastric cancer cells and lead to cell cycle arrest. Meanwhile, **9h** can induce Nur77 expression and nuclear export, suggesting that **9h** may mediate apoptosis through Nur77-Bcl-2 pathway. These findings prove that it is a feasible approach to develop anti-gastric cancer drugs targeting Nur77, and compound **9h** can be used as a lead compound for further studies.

## 2. Results

### 2.1. Chemistry

The general chemistry for the synthesis of 1-(2-(6-methoxynaphthalen-2-yl)-6-methylnicotinoyl)-4-substituted semicarbazide/thiosemicarbazide compounds **9a**–**9w** is outlined in Figure 2. (*E*)-3-(dimethylamino)-1-(6-methoxynaphthalen-2-yl)prop-2-en-1-one (**3**) was prepared by the reaction of 6-methoxy-2-acetonaphthone (**1**) with *N*-dimethylformamide dimethylacetal (DMF-DMA) (**2**), then refluxing of **3** with ethyl acetoacetate (**4**) in a mixture of acetic acid and ammonium acetate (molar ratio 8:10) afforded ethyl-6-(6-methoxynaphthalen-2-yl)-2-methyl nicotinate (**5**), the condensation of 5 with hydrazine hydrate (**6**) resulted in the formation of 6-(6-methoxynaphthalen-2-yl)-2-methylnicotinohydrazide (**7**), which was adapted from the method reported previously [24]. Finally, compound **7** was converted into the target compounds **9a**–**9w** by the reactions of **7** with treatment with the corresponding isocyanate/isothiocyanate (**8**). The target compounds were characterized by ^1^H-NMR, ^13^C-NMR and HRMS. Take compound **9h**, for example, the broad peaks at 8.38, 9.03 and 10.35 ppm correspond to three NH, two single peaks at 2.72 and 3.91 ppm correspond to methyl group on the pyridine ring and on the naphthalene ring, respectively. The remaining 12 hydrogens at 7.06–8.65 ppm correspond to hydrogen substituted on the benzene ring, pyridine ring and naphthalene ring. The calculated value of C_25_H_22_ClN_4_O_3_ [M + H]^+^ is 461.1375 and the measured value is 461.1787, the relative error is 2.6 ppm. Therefore, it can be proved that the structure of **9h** is accurate.

### 2.2. Biological Evaluation

#### 2.2.1. Antiproliferative Screening In Vitro

Antiproliferative activities of all the derivatives were assessed with a panel of three human cancer cell lines, namely MCF-7 (breast carcinoma), HGC-27 (gastric cancer) and HeLa (cervical cancer) (Table 1).

As shown in Table 1, it was observed that most of the compounds had inhibited growth of three tumor cell lines with moderate IC_50_ values. Results have shown that compounds **9h** and **9u** exhibited good cytotoxicities against HGC-27 cells with the IC_50_ of 1.40 μM and 4.56 μM. Among the compounds, **9h** and **9u** in particular displayed more potent anti-proliferative activity than the other compounds. We used compound **9h** and **9u** for further biological activity studies.

#### 2.2.2. Effect of Compound **9h** and **9u** on Cell Growth, Cell Cycle and Apoptosis

Compound **9h** and **9u** has the ability to inhibit HGC-27 cell growth by colonogenic survival assays. Treatment of cells with compound **9h** and **9u** for two weeks completely inhibited colony formation of HGC-27 cells in a dose-dependent manner (Figure 1). To rule out whether compound **9h** and **9u** played a role in cell cycle, flow cytometry was performed to examine cell cycle after 8 h of treatment. The results revealed that compound **9h** and **9u** induced cell cycle blockage in G_2_-M phase (Figure 2A). Production of cleaved PARP is an important index to assess whether apoptosis has happened. Compound **9h** and **9u** could induce the PARP cleavage in a dose-dependent and time-dependent manner (Figure 2B). Thus, by blocking the cell cycle at the G_2_-M phase, compound **9h** and **9u** led to cell growth inhibition and cell apoptosis in HGC-27. On the other hand, we assessed the toxicity of **9h** and **9u** on various cell lines including five cancer cell lines, HepG2, A549, MDA-MB-231, H460 and A875, and two normal cell lines, MRC-5 and LO2. The results showed that **9u** exhibited stronger toxicity toward normal cell lines than **9h** (Table 2). Therefore, we selected **9h** as the hit compound for further research.

#### 2.2.3. **9h** Induces Expression and Nuclear Export of Nur77, Which May Mediate **9h**-Induced Apoptosis

Several reports have confirmed that compounds can mediate apoptosis by regulating Nur77 expression and nuclear localization, especially by activating the classical apoptotic Nur77-Bcl2 signaling pathway [25]. We therefore examined the effect of **9h** on Nur77 expression and showed that **9h** induced Nur77 expression and PARP cleavage in a concentration-dependent manner (Figure 3A). Localization of Nur77 was determined by nuclear-cytosol fractionation experiments and immunofluorescence; the results showed that **9h** significantly induced Nur77 nuclear export (Figure 3B,C), indicating that **9h** may mediate apoptosis in gastric cancer cells mainly by regulating Nur77 expression and localization.

#### 2.2.4. Molecular Docking Study of **9h** and Nur77

We further performed molecular docking experiments to predict the interactions between **9h** and Nur77-LBD to identify a putative binding model for this molecule. Firstly, the native ligand **3mj** of the cocrystal structure was re-docked into Nur77-LBD (ligand-binding domain of Nur77) using the Induced Fit Docking (IFD) method to evaluate the docking performance. The overlay of these structures showed that the re-docked ligand was consistent with its pose in the crystal structures (RMSD value 0.98 Å, Figure 4). Next, we used the same IFD protocol to execute the molecular docking of **9h**. As shown in Figure 5A, **9h** could bind to the active pocket of Nur77, and the naphthalene group of which was inserted into the hydrophobic pocket of Nur77-LBD. Docking results depicted how **9h** interacted with the binding site of Nur77 (Figure 5B). The binding of **9h** to Nur77 is mainly contributed to by hydrogen bonding interactions and hydrophobic interactions. Table 3 described in detail the specific distances of hydrophobic interactions of hydrophobic moiety of **9h** with hydrophobic cavity formed by LEU118, LEU162, LEU165, VAL167, PHE172, LEU175, LEU178, VAL179, LEU224, LEU228 and LEU231. From Table 4, it was found that **9h** formed five hydrogen bonds with GLU114 (2.36 Å), VAL179 (2.54 Å and 2.53 Å) and ARG232 (1.85 Å and 3.39 Å), respectively, which would facilitate the stable binding of **9h** to Nur77. Besides, **9h** ($\Delta G_{bind}$ = −85.60 kcal/mol) exhibited higher binding affinity compared to the predicted binding free energy of the native ligand **3mj** ($\Delta G_{bind}$ = −62.41 kcal/mol), which was also consistent with their IFDScore (**9h** = −481.19, **3mj** = −475.23). Collectively, **9h** could be a potent potential Nur77 binder to exert anti-tumor activities.

## 3. Experimental Section

### 3.1. General Information

All reagents were purchased and used without further purification unless otherwise indicated. Reactions were magnetically stirred and monitored by thin-layer chromatography (TLC) on Merck silica gel 60F-254 (Darmstadt, Germany) under UV light. The final compounds were purified by column chromatography. ^1^H-NMR and ^13^C-NMR spectra were recorded on Bruker AV2 600 MHz (Billerica, MA, USA). Chemical shifts were given in parts per million (ppm) relative to tetramethylsilane (TMS) as an internal standard. High-resolution mass spectral (HRMS) data were acquired on a Q Exactive LC-MS/MS (Thermo Scientific, Waltham, MA, USA) instrument with UV detection at 254 nM in low-resonance electrospray mode (ESI). The molecular masses of compounds were calculated by CHEMDRAW ULTRA 12.0 Software (PerkinElmer Inc., Waltham, MA, USA).

### 3.2. Synthesis

#### 3.2.1. Procedure for Preparation of the (E)-3-(Dimethylamino)-1-(6-methoxynaphthalen-2-yl)prop-2-en-1-one (**3**)

The mixture of 6-methoxy-2-acetonaphthone (**1**) (4 g, 20 mmol) and *N*-dimethylformamide dimethylacetal (DMF-DMA, **2**) (2.64 mL, 20 mmol) was stirred at room temperature overnight, and the reaction was monitored by TLC. After the reaction was finished, the solvent was evaporated. The solid product **3** was recovered and recrystallized from ethanol as pure yellow powder (m.p. 125–127 °C; yield: 91%; ^1^H-NMR (600 MHz, DMSO-*d*_6_): δ 9.46 (s, 1H), 8.98–7.17(m, 6H), 4.90 (s, 1H), 4.36 (s, 3H), 4.08 (s, 6H)) [24].

#### 3.2.2. Procedure for Preparation of the Ethyl-6-(6-methoxynaphthalen-2-yl)-2-methylnicotinate (**5**)

To a solution of acetic acid and ammonium acetate (molar ratio 8:10), (750 mg, 3 mmol) of enaminone **3** and (400 mg, 3 mmol) of ethylacetoacetate (**4**) were added. The reaction mixture was then refluxed for 6 h and monitored by TLC. After the reaction was cooled down to room temperature, the precipitated white powder was filtered off and washed with ethanol to afford compound **5** (m.p. 112–114 °C; yield: 94%; ^1^H-NMR (600 MHz, DMSO-*d*_6_) δ 8.68 (s, 1H), 8.34-8.23 (m, 2H), 8.07 (d, *J* = 8.4 Hz, 1H), 8.00 (d, *J* = 9.0 Hz, 1H), 7.95 (d, *J* = 8.8 Hz, 1H), 7.40 (d, *J* = 2.2 Hz, 1H), 7.23 (dd, *J* = 2.4, 9.0 Hz, 1H), 4.35 (q, *J* = 7.0 Hz, 2H), 3.92 (s, 3H), 2.84 (s, 3H), 1.36 (t, *J* = 7.0 Hz, 3H) [24].

#### 3.2.3. Procedure for Preparation of 6-(6-Methoxynaphthalen-2-yl)-2-methylnicotinohydrazide (**7**)

The solution of ethyl-6-(6-methoxynaphthalen-2-yl)-2-methylnicotinate (**5**) and excess hydrazinehydrate 80% (**6**) in ethanol (10mL) was refluxed for 6 h at 80 °C. After the reaction was cooled down to room temperature, the resulted precipitate was filtered off, washed with ethanol and recrystallized from ethanol to give compound **7** as yellow powder (m.p. 220–222 °C; yield: 70%; ^1^H-NMR (400 MHz, DMSO-d_6_) δ 9.64 (s, 1H), 8.62 (d, *J* = 1.4 Hz, 1H), 8.24 (dd, *J* = 2.0, 8.6 Hz, 1H), 8.00–7.90 (m, 3H), 7.81 (d, *J* = 8.13 Hz, 1H), 7.37 (d, *J* = 2.5 Hz, 1H), 7.22 (dd, *J* = 2.5, 9.0 Hz, 1H), 4.57 (brs, 2H), 3.91 (s, 3H), 2.65 (s, 3H)) [24].

#### 3.2.4. Procedure for the Preparation of **9a**–**9w**

A solution of 6-(6-methoxynaphthalen-2yl)-2-methylnicotinohydrazide (**7**) and appropriate isocyanate/isothiocyanate (**8**, 0.05 mol) in ethanol (10 mL) was refluxed at 80 °C for about 4 h. The reaction was monitored by TLC. After completion of the reaction, the mixture was concentrated and the residue was purified by column chromatography (eluent: ethyl acetate/petroleum ether = 1/4) to obtain white solid. ^1^H-, ^13^C- spectra of all synthetized molecules are available in the Appendix A.

*4-Cyclopentyl-1-(2-(6-methoxynaphthalen-2-yl)-6-methylnicotinoyl)semicarbazide* (**9a**): m.p. 231–233 °C; yield: 60.1%; ^1^H-NMR (600 MHz, DMSO-d_6_) δ 9.99 (s, 1H), 8.64 (s, 1H), 8.25 (dd, *J* = 1.3, 8.6 Hz, 1H), 7.99 (t, *J* = 9.2 Hz, 2H), 7.93 (dd, *J* = 2.8, 8.3 Hz, 2H), 7.84 (s, 1H), 7.38 (d, *J* = 1.8 Hz, 1H), 7.22 (dd, *J* = 2.8, 8.3 Hz, 1H), 6.41 (d, *J* = 7.3 Hz, 1H), 3.98-3.92 (m, 1H), 3.91 (s, 3H), 2.70 (s, 3H), 1.88–1.77 (m, 2H), 1.70–1.60 (m, 2H), 1.56–1.47 (m, 2H), 1.40 (qd, *J* = 6.3, 12.2 Hz, 2H); ^13^C-NMR (150 MHz, DMSO-d_6_) δ 168.3, 158.5, 158.1, 156.6, 156.5, 137.1, 135.3, 133.6, 130.8, 128.9, 128.6, 127.6, 126.5, 125.2, 119.6, 117.3, 106.3, 55.7, 51.6, 33.1, 23.7, 23.6; ESI-HRMS (+): *m*/*z* calcd for C_24_H_27_N_4_O_3_ [M + H]^+^ 419.2078, found 419.2070.

*4-Vyclohexyl-1-(2-(6-methoxynaphthalen-2-yl)-6-methylnicotinoyl)semicarbazide* (**9b**): m.p. 236–237 °C; yield: 64.2%; ^1^H-NMR (600 MHz, DMSO-*d*6) δ 9.99 (s, 1H), 8.63 (s, 1H), 8.33–8.21 (m, 1H), 7.99 (t, *J* = 8.4 Hz, 2H), 7.95–7.91 (m, 2H), 7.86 (s, 1H), 7.38 (d, *J* = 1.8 Hz, 1H), 7.22 (dd, *J* = 2.4, 9.0 Hz, 1H), 6.31 (d, *J* = 8.0 Hz, 1H), 3.91 (s, 3H), 3.51–3.43 (m, 1H), 2.69 (s, 3H), 1.79 (d, *J* = 9.5 Hz, 2H), 1.72–1.62 (m, 2H), 1.59–1.50 (m, 1H), 1.34–1.24 (m, 2H), 1.23–1.10 (m, 3H); ^13^C-NMR (150 MHz, DMSO-d_6_) δ 168.3, 158.5, 157.8, 156.6, 156.5, 137.1, 135.3, 133.6, 130.8, 128.9, 128.6, 127.6, 126.5, 125.2, 119.6, 117.3, 106.3, 55.7, 48.6, 33.5, 25.7, 25.1, 23.6; ESI-HRMS (+): *m*/*z* calcd for C_25_H_29_N_4_O_3_ [M + H]^+^ 433.2234, found 433.2244.

*4-Benzyl-1-(2-(6-methoxynaphthalen-2-yl)-6-methylnicotinoyl)semicarbazide* (**9c**): m.p. 228–229 °C; yield: 60.2%; ^1^H-NMR (600 MHz, DMSO-d_6_) δ 10.13 (s, 1H), 8.66 (s, 1H), 8.32–8.25 (m, 1H), 8.17 (s, 1H), 8.02 (t, *J* =7.0 Hz, 2H), 7.99 (d, *J* = 9.0 Hz, 1H), 7.94 (d, *J* = 8.4 Hz, 1H), 7.38 (d, *J* = 1.5 Hz, 1H), 7.34 (d, *J* = 4.4 Hz, 4H), 7.26–7.19 (m, 2H), 7.14 (brs, 1H), 4.34 (d, *J* = 7.0 Hz, 2H), 3.91 (s, 3H), 2.73 (s, 3H); ^13^C-NMR (150 MHz, DMSO-d_6_) δ 168.5, 158.8, 158.5, 156.7, 156.7, 141.0, 137.3, 135.4, 133.6, 130.8, 128.9, 128.7, 128.6, 128.5, 128.4, 127.7, 127.4, 127.0, 126.5, 125.3, 119.6, 117.2, 106.3, 55.7, 43.2, 23.7; ESI-HRMS (+): *m*/*z* calcd for C_26_H_25_N_4_O_3_ [M + H]^+^ 441.1921, found 441.1904.

*1-(2-(6-Methoxynaphthalen-2-yl)-6-methylnicotinoyl)-4-o-tolylsemicarbazide* (**9d**): m.p. 230–232 °C; yield: 66.9%; ^1^H-NMR (600 MHz, DMSO-d_6_) δ 10.27 (s, 1H), 8.66 (s, 1H), 8.58 (brs, 1H), 8.27 (dd, *J* = 1.0, 8.6 Hz, 1H), 8.14 (s, 1H), 8.05–7.97 (m, 3H), 7.94 (d, *J* = 8.8 Hz, 1H), 7.73 (brs, 1H), 7.38 (d, *J* = 1.8 Hz, 1H), 7.23 (dd, J = 2.4, 9.0 Hz, 1H), 7.20 (d, *J* = 7.7 Hz, 1H), 7.17 (t, *J* = 7.9 Hz, 1H), 7.00 (t, *J* = 7.3 Hz, 1H), 3.91 (s, 3H), 2.74 (s, 3H), 2.27 (s, 3H);^13^C-NMR (150 MHz, DMSO-d_6_) δ 168.4, 158.5, 156.8, 156.5, 156.2, 137.6, 137.2, 135.4, 133.5, 130.8, 130.7, 128.9, 128.4, 127.7, 126.6, 126.5, 125.3, 125.3, 123.8, 123.8, 119.6, 117.3, 106.3, 55.7, 23.6, 18.3; ESI-HRMS (+): *m*/*z* calcd for C_26_H_25_N_4_O_3_ [M + H]^+^ 441.1921, found 441.1933.

*1-(2-(6-methoxynaphthalen-2-yl)-6-methylnicotinoyl)-4-m-tolylsemicarbazide* (**9e**): m.p. 220–222 °C; yield: 60.9%; ^1^H-NMR (600 MHz, DMSO-d_6_) δ 10.21 (s, 1H), 8.85 (s, 1H), 8.65 (s, 1H), 8.29–8.25 (m, 2H), 8.05–8.01 (m, 1H), 7.99 (d, *J* = 8.8 Hz, 2H), 7.94 (d, *J* = 8.8 Hz, 1H), 7.39–7.35 (m, 2H), 7.31 (d, *J* = 8.0 Hz, 1H), 7.23 (dd, *J* = 2.5, 9.0 Hz, 1H), 7.17 (t, *J* = 7.8 Hz, 1H), 6.81 (d, *J* = 7.3 Hz, 1H), 3.91 (s, 3H), 2.74 (s, 3H), 2.29 (s, 3H); ^13^C-NMR (150 MHz, DMSO-d_6_) δ 168.5, 158.6, 156.8, 156.5, 155.9, 140.0, 138.3, 137.2, 135.4, 133.6, 130.8, 129.0, 128.9, 128.5, 127.7, 126.5, 125.3, 119.6, 119.5, 117.3, 116.1, 106.3, 55.7, 23.6, 21.7; ESI-HRMS (+): *m*/*z* calcd for C_26_H_25_N_4_O_3_ [M + H]^+^ 441.1921, found 441.1922.

*1-(2-(6-Methoxynaphthalen-2-yl)-6-methylnicotinoyl)-4-p-tolylsemicarbazide* (**9f**): m.p. 224–225 °C; yield: 68.1%; ^1^H-NMR (600 MHz, DMSO-d_6_) δ 10.20 (s, 1H), 8.82 (brs, 1H), 8.65 (s, 1H), 8.27 (d, *J* = 8.4Hz, 1H), 8.24 (brs, 1H), 8.05–8.01 (m, 1H), 8.00–7.96 (m, 2H), 7.94 (d, *J* = 8.8 Hz, 1H), 7.40 (d, *J* = 8.8Hz, 3H), 7.23 (dd, *J* = 2.2, 8.8 Hz, 1H), 7.09 (d, *J* = 8.0 Hz, 2H), 3.91 (s, 3H), 2.73 (s, 3H), 2.25 (s, 3H); ^13^C-NMR (150 MHz, DMSO-d_6_) δ 168.5, 158.5, 156.7, 156.5, 137.5, 137.2, 135.4, 133.5, 131.2, 130.8, 129.6, 128.9, 128.5, 127.7, 126.5, 125.2, 119.6, 117.3, 106.3, 55.7, 23.6, 20.8; ESI-HRMS (+): *m*/*z* calcd for C_26_H_25_N_4_O_3_ [M + H]^+^ 441.1921, found 441.1922.

*1-(2-(6-Methoxynaphthalen-2-yl)-6-methylnicotinoyl)-4-(4-methoxyphenyl)semicarbazide* (**9g**): m.p. 214–217 °C; yield: 70.1%; ^1^H-NMR (600 MHz, DMSO-d_6_) δ 10.19 (d, *J* = 1.10 Hz, 1H), 8.75 (s, 1H), 8.66 (s, 1H), 8.27 (dd, *J* = 1.47, 8.80 Hz, 1H), 8.21 (s, 1H), 8.05–8.01 (m, 1H), 7.99 (d, *J* = 8.8 Hz, 2H), 7.94 (d, *J* = 8.8 Hz, 1H), 7.42 (d, *J* = 8.8 Hz, 2H), 7.38 (d, *J* = 2.2 Hz, 1H), 7.23 (dd, J = 2.6, 8.8 Hz, 1H), 6.88 (d, *J* = 8.8 Hz, 2H), 3.91 (s, 3H), 3.72 (s, 3H), 2.73 (s, 3H); ^13^C-NMR (150 MHz, DMSO-d_6_) δ 168.5, 158.5, 156.7, 156.6, 156.1, 155.0, 137.2, 135.4, 133.5, 133.1, 130.8, 128.9, 128.5, 127.7, 126.5, 125.2, 120.8, 119.6, 117.3, 114.3, 106.3, 55.7, 55.6, 23.7; ESI-HRMS (+): *m*/*z* calcd for C_26_H_25_N_4_O_4_ [M + H]^+^ 457.1870, found 457.1858.

*4-(2-Chlorophenyl)-1-(2-(6-methoxynaphthalen-2-yl)-6-methylnicotinoyl)semicarbazide* (**9h**): m.p. 221–222 °C; yield: 62.1%; ^1^H-NMR (600 MHz, DMSO-d_6_) δ 10.37 (brs, 1H), 9.06 (brs, 1H), 8.65 (s, 1H), 8.40 (brs, 1H), 8.27 (dd, *J* = 1.0, 8.8 Hz, 1H), 8.16 (d, *J* = 8.0 Hz, 1H), 8.05–8.01(m, 1H), 8.00–7.97 (m, 2H), 7.94 (d, *J* = 8.4 Hz, 1H), 7.48 (d, *J* = 8.0 Hz, 1H), 7.38 (d, *J* = 1.8 Hz, 1H), 7.32 (t, *J* = 7.5 Hz, 1H), 7.23 (dd, *J* = 2.2, 8.8 Hz, 1H), 7.06 (t, *J* = 7.3 Hz, 1H), 3.91 (s, 3H), 2.74 (s, 3H); ^13^C-NMR (150 MHz, DMSO-d_6_) δ 168.4, 158.6, 156.9, 156.4, 137.2, 136.9, 136.3, 135.4, 133.5, 130.8, 130.7, 129.7, 128.9, 128.1, 127.7, 127.6, 126.6, 126.4, 125.2, 119.6, 117.4, 106.3, 55.7, 22.2, 23.6; ESI-HRMS (+): *m*/*z* calcd for C_25_H_22_ClN_4_O_3_ [M + H]^+^ 461.1375, found 461.1387.

*4-(3,5-Dimethylphenyl)-1-(2-(6-methoxynaphthalen-2-yl)-6-methylnicotinoyl)semicarbazide* (**9i**): m.p. 230–231 °C; yield: 69.0%; ^1^H-NMR (600 MHz, DMSO-d_6_) δ 10.21 (s, 1H), 8.77 (brs, 1H), 8.66 (s, 1H), 8.28 (s, 1H), 8.29–8.24 (m, 1H), 8.26 (d, *J* = 2.6Hz, 1H), 8.04–8.01 (m, 1H), 7.99 (d, *J* = 8.8 Hz, 2H), 7.94 (d, *J* = 8.8 Hz, 1H), 7.38 (s, 1H), 7.23 (dd, *J* = 2.0, 9.0 Hz, 1H), 7.15 (s, 2H), 6.62 (s, 1H), 3.91 (s, 3H), 2.75 (s, 3H), 2.24 (s, 6H); ^13^C-NMR (150 MHz, DMSO-d_6_) δ 168.5, 158.5, 156.8, 156.5, 155.9, 139.9, 138.1, 137.2, 135.4, 133.6, 130.7, 128.9, 128.5, 127.7, 126.5, 125.3, 124.0, 119.6, 117.3, 116.7, 106.3, 55.7, 23.6, 21.6; ESI-HRMS (+): *m*/*z* calcd for C_27_H_27_N_4_O_3_ [M + H]^+^ 455.2078, found 455.2067.

*4-(3,5-Difluorophenyl)-1-(2-(6-methoxynaphthalen-2-yl)-6-methylnicotinoyl)semicarbazide* (**9j**): m.p. 224–225 °C; yield: 64.0%; ^1^H-NMR (600 MHz, DMSO-d_6_) δ 10.31 (s, 1H), 8.71 (brs, 1H), 8.65 (s, 1H), 8.58 (s, 1H), 8.31–8.24 (m, 1H), 8.06–7.91 (m, 5H), 7.38 (d, *J* = 1.8 Hz, 1H), 7.34–7.27 (m, 1H), 7.23 (dd, *J* = 2.2, 8.8Hz, 1H), 7.06 (t, *J* = 8.0Hz, 1H), 3.91 (s, 3H), 2.73 (s, 3H); ^13^C-NMR (150 MHz, DMSO-d_6_) δ 168.4, 158.5, 156.8, 156.5, 155.7, 137.2, 135.4, 133.5, 130.8, 128.9, 128.3, 127.7, 126.5, 125.2, 124.3(d, *J* = 24.0 Hz), 119.6, 117.3, 111.5(d, *J* = 22.5 Hz), 106.3, 104.3(d, *J* = 12.0 Hz), 104.2, 55.7, 23.6; ESI-HRMS (+): *m*/*z* calcd for C_25_H_21_F_2_N_4_O_3_ [M + H]^+^ 463.1576, found 463.1587.

*4-(3,4-Dichlorophenyl)-1-(2-(6-methoxynaphthalen-2-yl)-6-methylnicotinoyl)semicarbazide* (**9k**): m.p. 227–228 °C; yield: 60.0%; ^1^H-NMR (600 MHz, DMSO-d_6_) δ 10.26 (s, 1H), 9.28 (brs, 1H), 8.65 (s, 1H), 8.57 (brs, 1H), 8.27 (d, *J* = 8.4 Hz, 1H), 8.04–8.02 (m, 1H), 7.99 (d, *J* = 8.8Hz, 2H), 7.95–7.91 (m, 2H), 7.54–7.51 (m, 1H), 7.48 (brs, 1H), 7.38 (d, *J* = 1.8Hz, 1H), 7.23 (dd, *J* = 2.2, 8.8 Hz, 1H), 3.91 (s, 3H), 2.74 (s, 3H); ^13^C-NMR (150 MHz, DMSO-d_6_) δ 168.5, 158.5, 156.8, 156.6, 155.7, 140.5, 137.2, 135.4, 133.5, 131.4, 131.0, 130.8, 128.9, 128.3, 127.7, 126.6, 125.2, 123.7, 123.6, 120.1, 119.6, 119.1, 117.3, 106.3, 55.7, 23.7; ESI-HRMS (+): *m*/*z* calcd for C_25_H_22_Cl_2_N_4_O_3_ [M + H]^+^ 495.0985, found 495.0975.

*4-Hexyl-1-(2-(6-methoxynaphthalen-2-yl)-6-methylnicotinoyl)thiosemicarbazide* (**9l**): m.p. 195–196 °C; yield: 62.2%; ^1^H-NMR (600 MHz, DMSO-d_6_) δ 10.23 (brs, 1H), 9.32 (brs, 1H), 8.66 (s, 1H), 8.28 (d, *J* = 8.4 Hz, 1H), 8.12 (d, *J* = 7.7 Hz, 2H), 8.04 (d, *J* = 8.0 Hz, 1H), 7.99 (d, *J* = 8.8 Hz, 1H), 7.94 (d, *J* = 8.4 Hz, 1H), 7.39 (brs, 1H), 7.26–7.19 (m, 1H), 3.91 (s, 3H), 3.48 (d, *J* = 4.4 Hz, 2H), 2.70 (s, 3H), 1.54 (brs, 2H), 1.28 (brs, 6H), 0.96–0.77 (m, 3H); ^13^C-NMR (150 MHz, DMSO-d_6_) δ 167.9, 158.5, 156.8, 137.5, 135.4, 133.5, 130.8, 128.9, 127.6, 126.5, 125.3, 119.6, 117.1, 113.2, 110.2, 106.3, 55.7, 44.2, 31.6, 29.2, 26.4, 23.9, 22.6, 14.4; ESI-HRMS (+): *m*/*z* calcd for C_25_H_31_N_4_O_2_S [M + H]^+^ 451.2162, found 451.2150.

*4-Cyclohexyl-1-(2-(6-methoxynaphthalen-2-yl)-6-methylnicotinoyl)thiosemicarbazide* (**9m**): m.p. 213–214 °C; yield: 70.5%; ^1^H-NMR (600 MHz, DMSO-d_6_) δ 10.19 (brs, 1H), 9.29 (brs, 1H), 8.65 (s, 1H), 8.27 (dd, *J* = 1.3, 8.6 Hz, 1H), 8.09–8.06 (m, 1H), 8.05–8.01 (m, 1H), 7.99 (d, *J* = 9.1 Hz, 1H), 7.94 (d, *J* = 8.8 Hz, 1H), 7.78 (brs, 1H), 7.38 (d, *J* = 2.2 Hz, 1H), 7.23 (dd, *J* = 2.5, 8.8 Hz, 1H), 4.16 (brs, 1H), 3.91 (s, 3H), 2.71 (s, 3H), 1.85 (brs, 2H), 1.72 (d, *J* = 12.1 Hz, 2H), 1.59 (d, *J* = 12.5 Hz, 1H), 1.37–1.21(m, 4H), 1.12 (brs, 1H); ^13^C-NMR (150 MHz, DMSO-d_6_) δ 167.8, 167.8, 158.5, 156.8, 137.4, 135.4, 133.5, 130.8, 128.9, 127.7, 127.6, 126.6, 126.5, 125.3, 119.6, 117.1, 106.3, 55.7, 32.3, 32.2, 25.7, 25.3, 23.9; ESI-HRMS (+): *m*/*z* calcd for C_25_H_29_N_4_O_2_S [M + H]^+^ 449.2006, found 449.1993.

*4-Benzyl-1-(2-(6-methoxynaphthalen-2-yl)-6-methylnicotinoyl)thiosemicarbazide* (**9n**): m.p. 215–216 °C; yield: 67.5%; ^1^H-NMR (600 MHz, DMSO-d_6_) δ 10.39 (brs, 1H), 9.59 (brs, 1H), 8.73 (brs, 1H), 8.67 (s, 1H), 8.29 (d, *J* = 8.4 Hz, 1H), 8.18 (d, *J* = 6.6 Hz, 1H), 8.05 (d, *J* = 8.0 Hz, 1H), 7.99 (d, *J* = 8.8 Hz, 1H), 7.94 (d, *J* = 8.4 Hz, 1H), 7.42–7.30 (m, 5H), 7.28–7.18 (m, 2H), 4.84 (d, *J* = 5.1 Hz, 2H), 3.91 (s, 3H), 2.74 (s, 3H); ^13^C-NMR (150 MHz, DMSO-d_6_) δ 168.0, 158.6, 157.2, 156.9, 139.8, 137.7, 135.4, 133.5, 130.8, 128.9, 128.5, 127.7, 127.5, 127.1, 126.6, 125.3, 119.6, 117.1, 106.3, 55.7, 47.3, 24.0; ESI-HRMS (+): *m*/*z* calcd for C_26_H_25_N_4_O_2_S [M + H]^+^ 457.1693, found 457.1680.

*1-(2-(6-Methoxynaphthalen-2-yl)-6-methylnicotinoyl)-4-phenethylthiosemicarbazide* (**9o**): m.p. 212–213 °C; yield: 66.1%; ^1^H-NMR (600 MHz, DMSO-d_6_) δ 10.31 (brs, 1H), 9.49 (brs, 1H), 8.68 (s, 1H), 8.29 (d, *J* = 8.4 Hz, 1H), 8.25 (d, *J* = 8.0 Hz, 1H), 8.14 (d, *J* = 7.3 Hz, 1H), 8.06 (d, *J* = 8.0 Hz, 1H), 8.00 (d, *J* = 9.1 Hz, 1H), 7.95 (d, *J* = 8.4 Hz, 1H), 7.39 (d, *J* = 1.5 Hz, 1H), 7.34–7.30 (m, 2H), 7.29–7.26 (m, 2H), 7.25–7.20 (m, 2H), 3.91 (s, 3H), 3.73 (d, *J* = 5.1 Hz, 2H), 2.89 (t, *J* = 7.5 Hz, 2H), 2.73 (s, 3H); ^13^C-NMR (150 MHz, DMSO-d_6_) δ 168.0, 158.6, 157.1, 156.9, 139.7, 137.6, 135.4, 133.5, 130.8, 129.1, 128.9, 127.7, 126.6, 126.6, 125.3, 119.6, 117.1, 106.3, 55.7, 45.9, 35.4, 24.0; ESI-HRMS (+): *m*/*z* calcd for C_27_H_27_N_4_O_2_S [M + H]^+^ 471.1849, found 471.1839.

*1-(2-(6-Methoxynaphthalen-2-yl)-6-methylnicotinoyl)-4-phenylthiosemicarbazide* (**9p**): m.p. 182–183 °C; yield: 66.1%; ^1^H-NMR (400 MHz, DMSO-d_6_) δ 10.49 (brs, 1H), 9.79 (s, 1H), 8.67 (s, 1H), 8.28 (dd, *J* = 1.7, 8.7 Hz, 1H), 8.18 (brs, 1H), 8.05 (d, *J* = 8.0 Hz, 1H), 7.99 (d, *J* = 9.0 Hz, 1H), 7.95 (d, *J* = 8.8 Hz, 1H), 7.50 (brs, 2H), 7.42–7.33 (m, 3H), 7.27–7.14 (m, 2H), 3.91 (s, 3H), 2.75 (s, 3H); ^13^C-NMR (100 MHz, DMSO-d_6_) δ 167.9, 158.6, 156.8, 139.7, 137.7, 135.4, 133.5, 131.9, 128.9, 128.7, 128.6, 128.6, 127.7, 126.6, 125.7, 125.3, 119.6, 117.2, 106.3, 55.8, 24.0; ESI-HRMS (+): *m*/*z* calcd for C_25_H_23_N_4_O_2_S [M + H]^+^ 443.1536, found 443.1513.

*1-(2-(6-Methoxynaphthalen-2-yl)-6-methylnicotinoyl)-4-o-tolylthiosemicarbazide* (**9q**): m.p. 181–182 °C; yield: 66.1%; ^1^H-NMR (400 MHz, DMSO-*d*6) δ 10.49 (s, 1H), 9.71 (brs, 1H), 9.60 (s, 1H), 8.66 (s, 1H), 8.27 (dd, *J* = 1.5, 8.7 Hz, 1H), 8.22 (brs, 1H), 8.04 (d, *J* = 8.13 Hz, 1H), 7.99 (d, *J* = 9.1 Hz, 1H), 7.94 (d, *J* = 8.8 Hz, 1H), 7.39 (d, *J* = 2.4Hz, 1H), 7.32–7.13 (m, 5H), 3.91 (s, 3H), 2.73 (s, 3H); ^13^C-NMR (100 MHz, DMSO-d*6*) δ 168.1, 158.6, 157.1, 156.8, 138.5, 137.8, 135.4, 133.5, 130.8, 130.5, 129.4, 128.9, 127.7, 127.3, 126.6, 126.4, 126.3, 125.3, 119.6, 117.2, 106.3, 55.8, 23.8, 18.2; ESI-HRMS (+): *m*/*z* calcd for C_26_H_25_N_4_O_2_S [M + H]^+^ 457.1693, found 457.1700.

*1-(2-(6-Methoxynaphthalen-2-yl)-6-methylnicotinoyl)-4-m-tolylthiosemicarbazide* (**9r**): m.p. 179–181 °C; yield: 68.5%; ^1^H-NMR (600 MHz, DMSO-d_6_) δ 10.46 (brs, 1H), 9.75 (brs, 1H), 8.66 (s, 1H), 8.28 (d, *J* = 8.4 Hz, 1H), 8.20 (brs, 1H), 8.05 (d, *J* = 8.0 Hz, 1H), 7.99 (d, *J* = 9.1 Hz, 1H), 7.94 (d, *J* = 8.4 Hz, 1H), 7.39 (d, *J* = 2.2 Hz, 1H), 7.33–7.15 (m, 4H), 7.01 (d, *J* = 7.3 Hz, 1H), 3.91 (s, 3H), 2.74 (s, 3H), 2.32 (s, 3H); ^13^C-NMR (150 MHz, DMSO-d_6_) δ 174.3, 165.6, 158.5, 156.8, 147.8, 139.5, 137.7, 135.4, 134.1, 133.5, 130.8, 128.9, 127.7, 126.6, 125.3, 119.6, 117.1, 109.5, 106.3, 55.8, 40.5, 21.5; ESI-HRMS (+): *m*/*z* calcd for C_26_H_25_N_4_O_2_S [M + H]^+^ 457.1693, found 457.1704.

*1-(2-(6-Methoxynaphthalen-2-yl)-6-methylnicotinoyl)-4-p-tolylthiosemicarbazide* (**9s**): m.p. 199–201 °C; yield: 68.5%; ^1^H-NMR (400 MHz, DMSO-d_6_) δ 10.47 (brs, 1H), 9.81 (brs, 1H), 9.72 (brs, 1H), 8.67 (s, 1H), 8.28 (dd, *J* = 1.4, 8.6Hz, 1H), 8.18 (brs, 1H), 8.05 (d, *J* = 8.1 Hz, 1H), 7.99 (d, *J* = 9.0 Hz, 1H), 7.94 (d, *J* = 8.7Hz, 1H), 7.41–7.32 (m, 3H), 7.23 (dd, *J* = 2.5, 8.8 Hz, 1H), 7.19 (s, 1H), 7.17 (s, 1H), 3.91 (s, 3H), 2.75 (s, 3H), 2.31 (s, 3H); ^13^C-NMR (100 MHz, DMSO-d_6_) δ 167.9, 158.6, 157.1, 156.8, 137.7, 137.1, 135.4, 134.7, 133.5, 130.8, 129.2, 129.1, 127.7, 126.6, 125.3, 119.6, 117.1, 106.3, 55.8, 24.0, 21.0; ESI-HRMS (+): *m*/*z* calcd for C_26_H_25_N_4_O_2_S [M + H]^+^ 457.1693, found 457.1706.

*1-(2-(6-Methoxynaphthalen-2-yl)-6-methylnicotinoyl)-4-(4-methoxyphenyl)thiosemicarbazide* (**9t**): m.p. 191–192 °C; yield: 64.0% °C ^1^H-NMR (400 MHz, DMSO-d_6_) δ 10.47 (brs, 1H), 9.77 (brs, 1H), 9.70 (brs, 1H), 8.67 (s, 1H), 8.29 (d, *J* = 7.7 Hz, 1H), 8.21 (brs, 1H), 8.05 (d, *J* = 8.1 Hz, 1H), 7.99 (d, *J* = 9.0 Hz, 1H), 7.94 (d, *J* = 8.8 Hz, 1H), 7.41–7.32 (m, 3H), 7.23 (dd, *J* = 2.3, 8.8 Hz, 1H), 6.95 (d, *J* = 8.8 Hz, 2H), 3.91 (s, 3H), 3.77 (s, 3H), 2.77 (s, 3H); ^13^C-NMR (100 MHz, DMSO-d_6_) δ 168.0, 158.6, 157.4, 157.1, 156.9, 137.7, 135.4, 133.6, 132.5, 130.8, 128.9, 127.7, 126.6, 125.3, 122.3, 119.6, 117.1, 113.9, 106.3, 55.8, 55.7, 24.0; ESI-HRMS (+): *m*/*z* calcd for C_26_H_25_N_4_O_3_S [M + H]^+^ 473.1642, found 473.1680.

*4-(2-Chlorophenyl)-1-(2-(6-methoxynaphthalen-2-yl)-6-methylnicotinoyl)thiosemicarbazide* (**9u**): m.p. 269–270 °C; yield: 68.0% °C ^1^H-NMR (400 MHz, DMSO-d_6_) δ 10.56 (brs, 1H), 9.94 (brs, 1H), 9.66 (s, 1H), 8.66 (s, 1H), 8.28 (dd, *J* = 1.3, 8.8 Hz, 1H), 8.24 (d, *J* = 4.4 Hz, 1H), 8.05 (d, *J* = 8.0 Hz, 1H), 7.99 (d, *J* = 9.0 Hz, 1H), 7.94 (d, *J* = 8.8 Hz, 1H), 7.53 (dd, *J* = 1.2, 7.8 Hz, 1H), 7.48 (brs, 1H), 7.42–7.35 (m, 2H), 7.34–7.28 (m, 1H), 7.23 (dd, *J* = 2.5, 8.9 Hz, 1H), 3.91 (s, 3H), 2.74 (s, 3H); ^13^C-NMR (100 MHz, DMSO-d_6_) δ 168.1, 158.6, 158.6, 157.2, 156.9, 137.8, 137.3, 135.4, 133.5, 131.9, 131.5, 131.1, 130.8, 129.8, 128.9, 128.6, 127.7, 126.6, 125.3, 119.6, 117.1, 106.3, 55.8, 23.9; ESI-HRMS (+): *m*/*z* calcd for C_25_H_22_ClN_4_O_2_S [M + H]^+^ 477.1147, found 477.1159.

*4-(2,4-Difluorophenyl)-1-(2-(6-methoxynaphthalen-2-yl)-6-methylnicotinoyl)thiosemicarbazide* (**9v**): m.p. 264–266 °C; yield: 68.0% °C ^1^H-NMR (600 MHz, DMSO-d_6_) δ 10.58 (brs, 1H), 10.04 (brs, 1H), 9.63 (brs, 1H), 8.67 (brs, 1H), 8.29 (d, *J* = 8.0 Hz, 1H), 8.25 (d, *J* = 6.2 Hz, 1H), 8.06 (d, *J* = 7.3 Hz, 1H), 7.99 (d, *J* = 8.8 Hz, 1H), 7.94 (d, *J* = 8.8 Hz, 1H), 7.43–7.37 (m, 2H), 7.34 (t, *J* = 8.2 Hz, 1H), 7.23 (dd, *J* = 2.4, 8.99 Hz, 1H), 7.12 (t, *J* = 7.7 Hz, 1H), 3.91 (s, 3H), 2.75 (s, 3H); ^13^C-NMR (150MHz, DMSO-d_6_) δ 183.0, 168.0, 158.6, 157.2, 156.9, 137.8, 135.4, 133.5, 132.5, 132.4(d, *J* = 21.0 Hz), 130.8, 128.9, 127.7, 126.6, 126.4, 125.3, 124.5(t, *J* = 19.5 Hz), 119.6, 117.1, 111.4(d, *J* = 27.0Hz), 106.3, 104.7, 104.7, 55.7, 24.0; ESI-HRMS (+): *m*/*z* calcd for C_25_H_21_F_2_N_4_O_2_S [M + H]^+^ 479.1348, found 479.1336.

*4-(3,4-Dichlorophenyl)-1-(2-(6-methoxynaphthalen-2-yl)-6-methylnicotinoyl)thiosemicarbazide* (**9w**): m.p. 219–222 °C; yield: 70.0%°C ^1^H-NMR (600 MHz, DMSO-d_6_) δ 10.54 (brs, 1H), 10.09 (brs, 1H), 9.97 (brs, 1H), 8.67 (brs, 1H), 8.29 (d, *J* = 8.4 Hz, 1H), 8.22 (brs, 1H), 8.07 (d, *J* = 7.7 Hz, 1H), 7.99 (d, *J* = 8.8 Hz, 1H), 7.94 (d, *J* = 8.8 Hz, 1H), 7.89 (brs, 1H), 7.65–7.61 (m, 1H), 7.59–7.55 (m, 1H), 7.38 (d, *J* = 1.8 Hz, 1H), 7.23 (dd, *J* = 2.3, 8.8 Hz, 1H), 3.91 (s, 3H), 2.76 (s, 3H); ^13^C-NMR (150 MHz, DMSO-d_6_) δ 168.0, 158.6, 157.0, 143.5, 140.8, 139.9, 137.7, 135.4, 135.1, 133.5, 130.8, 130.3, 128.9, 127.7, 127.5, 126.7, 126.6, 126.2, 125.3, 120.7, 119.6, 117.2, 106.3, 55.7, 24.1; ESI-HRMS (+): *m*/*z* calcd for C_25_H_21_Cl_2_N_4_O_2_S [M + H]^+^ 511.0757, found 511.0741.

### 3.3. Cell Culture

All of the cell lines were obtained from the American Type Culture Collection (ATCC, Manassas, VA, USA) and grown in DMEM or RPMI culture medium containing 10% fetal bovine serum (*v*/*v*) in 5% CO_2_ at 37 °C.

### 3.4. Cytotoxicity against Cell Lines

Confluent cells in good state were cultured in 96-well plates (5–10 × 10^4^ cells/mL). After cells were attached to the plate, compounds with various concentrations were applied at 37 °C for 72 h. Then, the cells were incubated with 10 μL of 5 mg/mL MTT reagent at 37 °C for 4 h. The supernatant was removed, and the cells were dissolved in 100 μL dimethyl sulfoxide and shaken for 5 min. Finally, the light absorption (OD) of the dissolved cells was measured at 490 nM.

### 3.5. Western Blot Analysis

Equal amounts of the lysates were electrophoresed on 10% SDS-PAGE gel and transferred onto PVDF membranes. After blocked with 5% nonfat milk in TBST (50 mM Tris-HCl (pH 7.4), 150 mM NaCl and 0.1% Tween 20) for 1 h, the membranes were incubated with various primary antibodies overnight and secondary antibodies for 3 h, finally detected using an ECL system. The blots were captured with ChemiDoc XRS+System (Bio-Rad, Hercules, CA, USA).

### 3.6. Flow Cytometry Assay

Flow cytometry analysis was applied for cell cycle detection. Firstly, HGC-27 cell was adjusted to 1 × 10^6^/mL, inoculated into a six-well plate and placed in an incubator at 37 °C containing 5% CO_2_ saturated humidity overnight. After the cells were fully attached to the plate, 10 mM **9u** and **9h** were administrated to the cells for 8 h. After cells were collected and fixed with 70% ethanol, DAPI was applied to stain the cells at the final concentration of 1 μg/mL; subsequently, on-board testing by flow cytometry (Cytoflex S (Beckman Coulter, Brea, CA, USA)) was conducted. 

### 3.7. Colony Formation Assay

HGC-27 cells were cultured in a six-well plate (500 cells/well) and treated with **9u** and **9h** in 10% serum medium for two weeks, then fixed with methanol at room temperature and stained with 0.1% crystal violet, and images were scanned with a scanner.

### 3.8. Nuclear-Cytosol Fractionation Assay

Cells in the logarithmic phase of growth were seeded in 6-cm dishes; after, apposition cells were treated with compound **9h** for 3 h. The medium was discarded and washed twice with pre-chilled PBS, then the cells were scraped off with a cytoscraper and collected into 1.5 mL centrifuge tubes and centrifuged at 1500 rpm for 3 min. Next, 80 μL of Buffer B (containing 100× protease inhibitor) were added for 20 min in an ice bath, followed by 20 μL of Buffer B containing 1% TritonX-100. The mixture was vortexed for 10 s, centrifuged at 10,000× *g* for 15 min at 4 °C, the supernatant collected as cytoplasmic proteins and the precipitate as the nuclear fraction. Then, 100 μL of Buffer C containing 0.4% TritonX-100 were added and centrifuged for 15 min at 4 °C, and 100 uL of Buffer C containing 0.4% TritonX-100 were added and vortexed for 30 min at 4 °C, centrifuged at 15,000× *g* for 15 min at 4 °C, and the supernatant was the nucleus protein.

### 3.9. Immunofluorescence

The cells were inoculated into the culture dish with preplaced treated coverslips, and the coverslips were removed when the cells were close to growing into a monolayer. Then, they were washed twice with PBS and 4% paraformaldehyde was used to fix the cells for 30 min. The cells were permeabilized with 1‰ TritonX-100 for 10 min. After permeabilization, the cells were washed with PBS for 3 × 5 min. Then, the cells were blocked with 5% BSA blocking solution for 30 min. The primary antibody was incubated for 1 h at room temperature. The cells were rinsed three times with PBST, each rinse for 5 min. The cells were incubated at room temperature in the dark for 1 h. The cells were rinsed three times with PBST, each rinse for 5 min, followed by one distilled water rinse. One drop of blocker containing DAPI dye was added and the slice sealed and this was examined by fluorescence microscopy.

### 3.10. The Docking Experiments

The cocrystal structure of human Nur77-LBD was downloaded from the Protein Data Bank website (https://www.pdbus.org/structure/4RE8, accessed on 2 February 2022) [25]. Schrödinger (Version 2019-1, Schrödinger, LLC, New York, NY, USA) was used to carry out molecular docking experiments. The crystal structure of Nur77-LBD was modified and prepared by the Protein Preparation Wizard module of Schrödinger with the default settings, and the 2D structure of target molecule **9h** was converted to a 3D structure and prepared by the LigPrep module with default parameters. An OPLS3e [25] force field was employed for both protein and ligands minimization. All of the docking calculations were conducted by the standard protocol of Induced Fit Docking [26,27,28] without changing any settings. In addition, the MM/GBSA module (Prime MMGBSA v3.000, Schrödinger, LLC, New York, NY, USA) was accessed to calculate the binding free energy under an implicit solvent model. The top-ranked docking score, docking orientation and binding energy were used for a comprehensive evaluation of the optimum docking conformation. The hydrophobic and hydrogen bonding interactions between **9h** and Nur77 were identified by Protein-Ligand Interaction Profiler (PLIP) web tool [29]. Maestro (Schrödinger, LLC, New York, NY, USA) [30], Schrödinger’s graphical user interface, was mainly deployed for the visualization of docking study. The figures of the molecular modeling were prepared with PyMOL (Version 2.3 Schrödinger, LLC, New York, NY, USA) [31].

## 4. Conclusions

In this paper, a new series of 1-(2-(6-methoxynaphthalen-2-yl)-6-methylnicotinoyl)-4-substituted semicarbazide/thiosemicarbazide derivatives **9a**–**9w** was synthesized and their biological evaluation as potential anticancer agents was conducted. The synthetic method was relatively simple, and the compounds were easily purified and produced in high yields. Our results suggest that 1-(2-(6-methoxynaphthalen-2-yl)-6-methylnicotinoyl)-4-substituted semicarbazide/thiosemicarbazide derivatives have good anti-tumor effects on several cancer cells, especially on gastric cancer cell HGC-27. Compound 9h probably mediates apoptosis mainly through regulation of Nur77 and may be a promising anti-tumor leading compound for further research.

## Data Availability

Available from authors.

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
