# Peer review of "Synthesis and Biological Evaluation of 1-(2-(6-Methoxynaphthalen-2-yl)-6-methylnicotinoyl)-4-Substituted Semicarbazides/Thiosemicarbazides as Anti-Tumor Nur77 Modulators"

_molecules, 2022, doi:10.3390/molecules27051698_

Round 1
Reviewer 1 Report
The manuscript entitled “Synthesis and biological evaluation of 1-(2-(6-methoxynaphthalen-2-yl)-6-methylnicotinoyl)-4-substituted semicarbazides/thiosemicarbazides as anti-tumor Nur77 modulators” by Hongyu Hu et al. described the synthesis of a new series of 1-(2-(6-methoxynaphthalen-2-yl)-6-methylnicotinoyl)-4-substituted semicarbazide/thiosemicarbazide derivatives and their evaluation as potential anticancer agents against various cancer cell lines such as A549, HepG2, HGC-27, MCF-7 and HeLa. The manuscript may be of general interest to the researchers of this field, but the manuscript lacks some information that the author should consider and incorporate in the present form of the manuscript. Here are a few concerns that need to be addressed in the present form of the manuscript.
- The background addressed in a broad context, highlight the purpose of the study and briefly description of the main synthetic method applied should be added in the abstract.
- The keyword " thiosemicarbazid" should be “thiosemicarbazide”
- Scheme 2 should be corrected: compounds 8 “RNCO/S” and compound 9 “…=O/S” should be corrected as “RNCX” and “….=X” with note “X = S, O”; R should be added at the bottom of the scheme 2.
- Interpretation of spectral data for synthesized compounds 9a-9w should be discussed in the text “Results, 2.1. Chemistry”.
- Chemical names of compounds 9a-9w in the experimental part should be in capital letters.
Reviewer 2 Report
The manuscript “Synthesis and biological evaluation of 1-(2-(6-methoxynaphtha- 2len-2-yl)-6-methylnicotinoyl)-4-substituted semicarbazides/thi- 3” osemicarbazides as anti-tumor Nur77 modulators” by Hongyu Hu et al. focuses on the antitumor activity of newly synthesized compounds. The manuscript is of general interest and will target a large audience. The experimental choice is adequate but some of the execution has to be ascertained. The English language needs a lot of improvements.
The main issues are as follows.
From the series of compounds the authors have discerned one 9h. However, 9h the purity is questionable. It has been assessed by NMR and some unassigned (9.75 and 7.80 ppm etc. ..) suggest 10-15% impurities.
The MTT as described on P13l422 3.4 is been reported for 490 nm absorption. This need and explanation as usually 570 + nm is used . (NB! 490 nm may be fine but RPMI and DMEM )
The IC50 are quite high and the use of concentration in the µm range are not really favored as undetected effects are plausible.
The recommendation is for major revisions.
Minor
Please rephrase (abstract) :
Some of these compounds, especially 9h showed the most potent anti-proliferative activity.
Suggestion: From the series of compounds 9h exhibited the most potent anti-proliferative activity against ….
P1.Line18 “Cloney formation” . Colony formation?
“Mechanically “- not correct?
P2L45 However, currently synthesized small molecule com- pounds targeting Nur77 are not strong enough in targeting, activity and specificity.
Targeting and targeting. . Is the second “targeting” synonym of binding e.g. Ka? Please clarify or rephrase.
P2 L54 :” binocyclic aromatic rings” . Not sure about binocyclic term. Please check.
Scheme 1. Captions should be corrected to antitumor agents containing Urea/ thiourea moiety .
Scheme 2. The C=O of the ethyl acetoacetate is to be corrected
Table 1 . What are the units of the IC50 ? Are you sure about the use of CisPt as control ?
Why flow cytometry is conducted after 8h. The cell cycle ( if synchronized) is ~ 23 h? If not synchronized ….
What are those reported hydrophobic interactions? Are you talking about C-H…pi interactions and pi…pi, weak C-H…O etc…. ?
Table 4. The last 232ARG with D…A of 3.96 angstr is not consistent with Hydrogen bonding . Please check/remove.
P13L415 CO2 the 2 subscript.
Round 2
Reviewer 2 Report
The manuscript has been significantly improved. The investigation work on 9h is now fine.
The docking, although not an issue, is a minor concern. It is nice to use directly PLIP server settings but usually HB threshold is 3.6-3.7 Angs ( speaking generously). The use of 4.1 Angs cutoff is a bit excessive, and probably intentionally left in the PLIP.
Regards,